# Dansgaard-Oeschger cycles of the penultimate and last glacial period recorded in stalagmites from Türkiye

F. Held [1] ✉, H. Cheng [2], R. L. Edwards[3], O. Tüysüz[4], K. Koç [5] & D. Fleitmann [1] ✉

The last glacial period is characterized by abrupt climate oscillations, also known as Dansgaard-Oeschger (D-O) cycles. However, D-O cycles remain poorly documented in climate proxy records covering the penultimate glacial period. Here we present highly resolved and precisely dated speleothem time series from Sofular Cave in northern Türkiye to provide clear evidence for D-O cycles during Marine Isotope Stage (MIS) 6 as well as MIS 2-4. D-O cycles are most clearly expressed in the Sofular carbon isotope time series, which correlate inversely with regional sea surface temperature (SST) records from the Black Sea. The pacing of D-O cycles is almost twice as long during MIS 6 compared to MIS 2-4, and could be related to a weaker Atlantic Meridional Overturning Circulation (AMOC) and a different mean climate during MIS 6 compared to MIS 2-4, leading most likely to a higher threshold for the occurrence of D-O cycles.

The last glacial period is characterized by centennial- to millennial-scale climate oscillations known as Dansgaard - Oeschger (D-O) cycles. At least 25 D-O cycles were identified in Greenland ice cores and characterized by warm Greenland interstadials (GI) and cold Greenland stadials (GS)[1–3]. Though the timing, duration, and spatial extent of D-O cycles are well-documented, uncertainties exist regarding their triggers and pacing. Proposed causes for D-O cycles include changes in the Atlantic Meridional Overturning Circulation (AMOC) and solar and volcanic forcing[4–9]. While D-O cycles are well documented for the last glacial period[1,10–15], there are only very few terrestrial proxy records which provide unambiguous evidence for their existence during the preceding glacial periods[12–14,16–19]. This knowledge gap limits our ability to evaluate the influence of different glacial boundary conditions on D-O variability, which would contribute to a better understanding of their driving mechanisms. The penultimate glacial period corresponding to Marine Isotope Stage (MIS) 6 (191-130 kyrs BP) was most likely one of the strongest Quaternary glaciations, with much greater ice sheet extent throughout Eurasia[20,21]. Though MIS 6 is not directly covered by Greenland ice core records, a synthetic δ[18]O Greenland record suggests the occurrence of D-O like cycles during the penultimate glacial[22]. Further records are needed to better understand D-O cycles and to characterize their pattern, timing, and spatial extent. Such information can be obtained from speleothems, which are known to be an ideal archive for identifying D-O cycles[15]. However, speleothem-based climate reconstructions covering the penultimate and preceding glacial periods are still rare and most of them do not display a clear D-O pattern in their isotopic profiles[11,23–29]. At present, D-O cycles have been explicitly identified in tropical and subtropical speleothems from Sanbao Cave[16] and Huagapo Cave[17], respectively, but additional precisely dated and highly resolved paleoclimate records are urgently needed to verify the synthetic Greenland ice core record and to develop a more detailed stratigraphy for D-O cycles during MIS 6.

Here we present an extended Sofular record for MIS 2-4 and parts of MIS 6/7 to show D-O cycles in very close detail. Complementing previous studies[30,31], we present so far unpublished δ[13]C isotope profiles to

[1]Department of Environmental Sciences, University of Basel, 4056 Basel, Switzerland. [2]Institute of Global Environmental Change, Xi'an Jiaotong University, 710054 Xi'an, China. [3]Department of Earth and Environmental Sciences, University of Minnesota, 55455 Minneapolis, USA. [4]Eurasia Institute of Earth Sciences, Istanbul Technical University, 34469 Istanbul, Türkiye. [5]Department of Geological Engineering, Akdeniz University, 07058 Antalya, Türkiye. ✉e-mail: frederick.held@unibas.ch; dominik.fleitmann@unibas.ch

provide further evidence for a very close atmospheric coupling between the eastern Mediterranean and the North Atlantic during the penultimate glacial period. Because of the very strong atmospheric connection between the North Atlantic region and Northern Türkiye[15,30,31], Sofular stalagmites are very well suited to investigate millennial-scale variability for glacial periods preceding MIS 2-4.

## Results and discussion
### Study site
Sofular Cave (41°25′N, 31°56′E) is located in the North Anatolian mountains in Türkiye, ~10 km inland from the southern Black Sea coast (Fig. 1). The cave is within the Lower Cretaceous limestones. Mean annual temperature and total precipitation average ~13.3 °C and ~1,200 mm yr$^{-1}$, respectively. Although ~75% of total annual precipitation occurs between September and April, infiltration of rainwater occurs most likely throughout the year. Moisture originates predominantly from the Black Sea and, to a lesser extent, from the Marmara and Mediterranean Sea[30,31]. The soil above the cave is of heterogeneous thickness and covered by dense vegetation of deciduous trees (e.g., oaks, elms, and beeches), shrubs, and marginal grasslands.

### Chronology
The chronology of the Sofular record is based on 150 $^{230}$Th ages, all of which are in stratigraphic order. Uranium concentrations of ~0.5 ppm and low common thorium result in very precise $^{230}$Th ages for all stalagmites with typical age uncertainties of ~0.5%. The Sofular record covers the last 205,000 years before present (BP = 1950) almost continuously, except for two gaps between 24.1 - 21.6 kyrs BP and 160 - 134 kyrs BP (Figs. 2 and 3).

### Sofular isotope profiles
The average temporal resolution of the entire Sofular record is around 40 years. Including previous studies[30,31], 8540 stable oxygen (δ$^{18}$O) and

carbon (δ$^{13}$C) isotope measurements were performed. Additional high-resolution data were obtained from stalagmites So-4, So-13 and So-57, and we now present the complete Sofular δ$^{13}$C record. δ$^{18}$O and δ$^{13}$C values range from −7.6‰ to −17.8‰ (VPBD) and −2.7‰ to −10.6‰ (VPBD), respectively (Fig. 2). Oxygen and carbon isotope profiles of all stalagmites are very similar, with correlation coefficients (r) of stalagmites covering MIS 1-4 (So-1, So-2, So-4, So-13) typically ranging from -0.4 to -1.0 (determined using iscam[32])(Supplementary Fig. 1a, b). For MIS 6 (So-4 and So-57) correlation coefficients range from -0.7 to -1.0 (Supplementary Fig. 1c).

For both MIS 2-4 and MIS 6, isotope profiles show distinct centennial- to millennial-scale variability and D-O like patterns, which are, even on multi-decadal timescales, very similar to those recorded in Greenland ice cores (Fig. 3). The strong resemblance between the Sofular and NGRIP isotope records demonstrates the very close and rapid atmospheric coupling between the North Atlantic and Black Sea region[15,30,33].

D-O related variations of 0.5 to 1.5‰ in speleothem δ$^{18}$O are most likely caused by multiple factors such as air temperature, variations in δ$^{18}$O of the moisture source, seasonality, and amount of precipitation above Sofular Cave[30,31]. On orbital time-scales fluctuations in Sofular δ$^{18}$O values track changes in the oxygen isotopic composition of Black Sea surface water through the so-called water vapor source effect[30,31]. δ$^{18}$O values of around −8.5 ± 1‰ mark periods when the Black Sea was connected with the Mediterranean Sea when the sea level was higher than the Bosporus sill depth of ~35 meters below current sea level[31]. More negative δ$^{18}$O values are therefore indicative of a Black Sea "Lake"[34] without inflow of saline Mediterranean water. Pronounced negative excursions in δ$^{18}$O of up to −17.8‰, however, indicate the inflow of isotopically depleted meltwater from the Caspian Sea and rivers entering the Black Sea"Lake"[34] (Supplementary Fig. 2). Thus, the water vapor source effect in combination with variable mixing times of the Black Sea[35] affected D-O related fluctuations in δ$^{18}$O during MIS 2-4

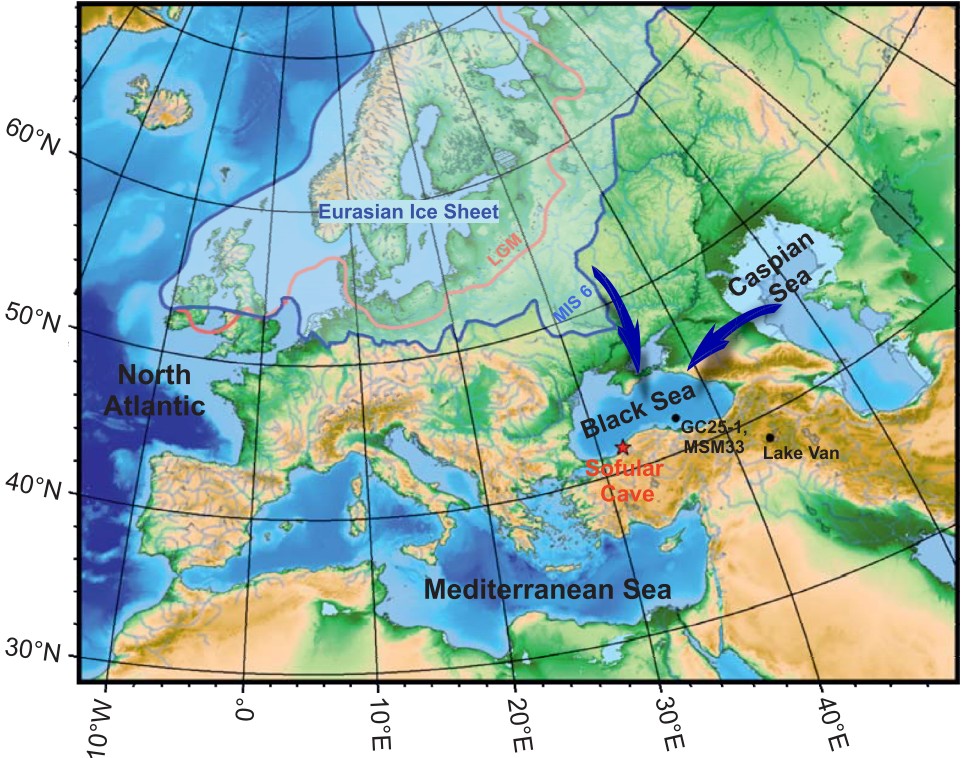

**Fig. 1 | Map showing Eurasia including the location of Sofular Cave (red star).** Black circles mark the location of Lake Van and Black Sea sediment cores GC25-1 and MSM33. The maximum extent of the Eurasian Ice Sheet during the Saalian (Marine Isotope Stage (MIS) 6; ice sheet boundaries redrawn after Svendsen et al.[20] and Last Glacial Maximum (LGM)[60] are shown by the blue and red lines, respectively. Arrows indicate potential meltwater pathways during MIS 6.

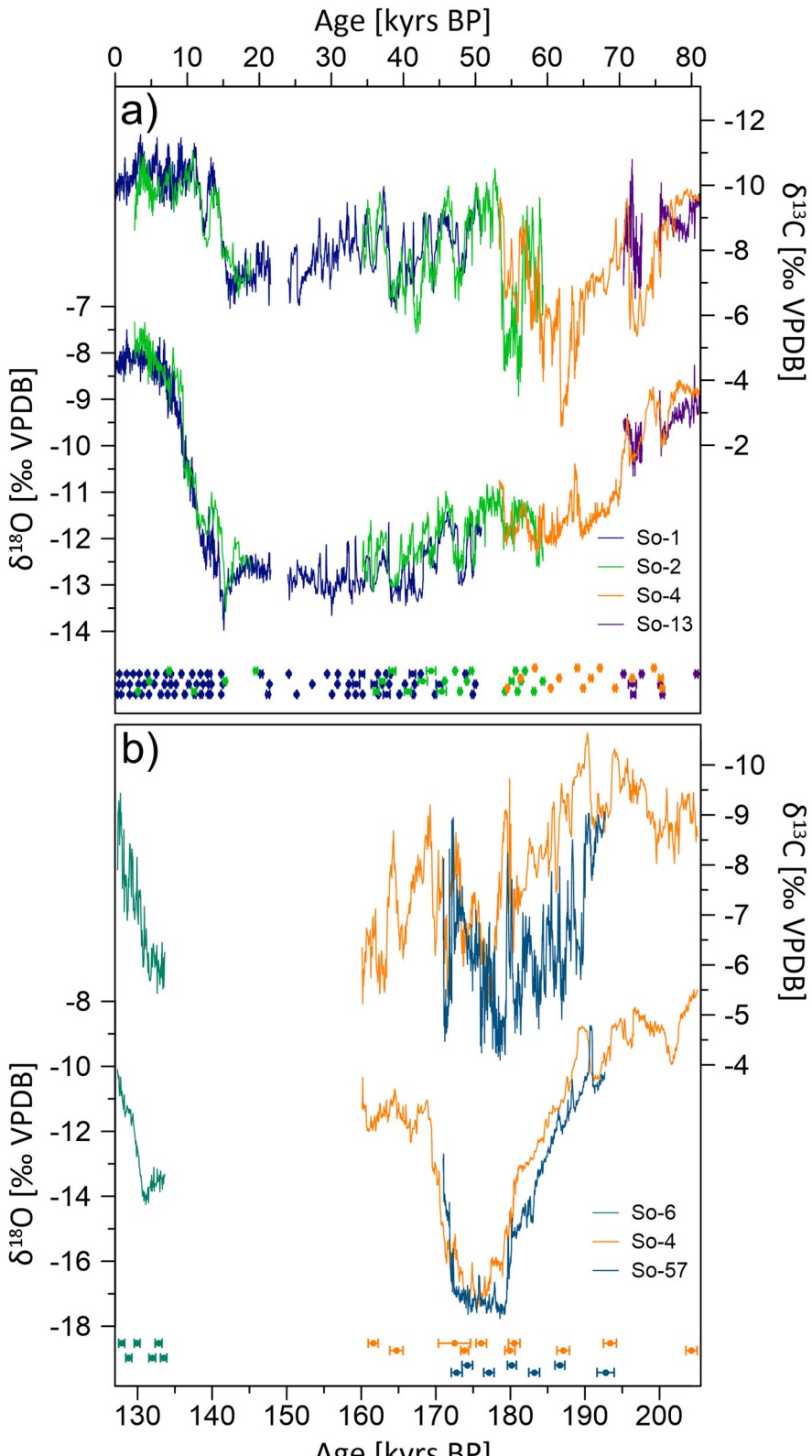

**Fig. 2 | Sofular stalagmite δ¹⁸O and δ¹³C records. a** Sofular stalagmites So-1, So-2, So-4 and So-13 covering the Holocene and last glacial period (data for So-1 and So-2 taken from Fleitmann et al.[30]). **b** Sofular stalagmites So-4, So-6 and So-57 covering the penultimate glacial period (δ¹⁸O data for So-4 and So-6 taken from Badertscher et al.[31]). Color coded ²³⁰Th ages with 2σ-error bars are plotted below. Source data are provided in the Source Data file.

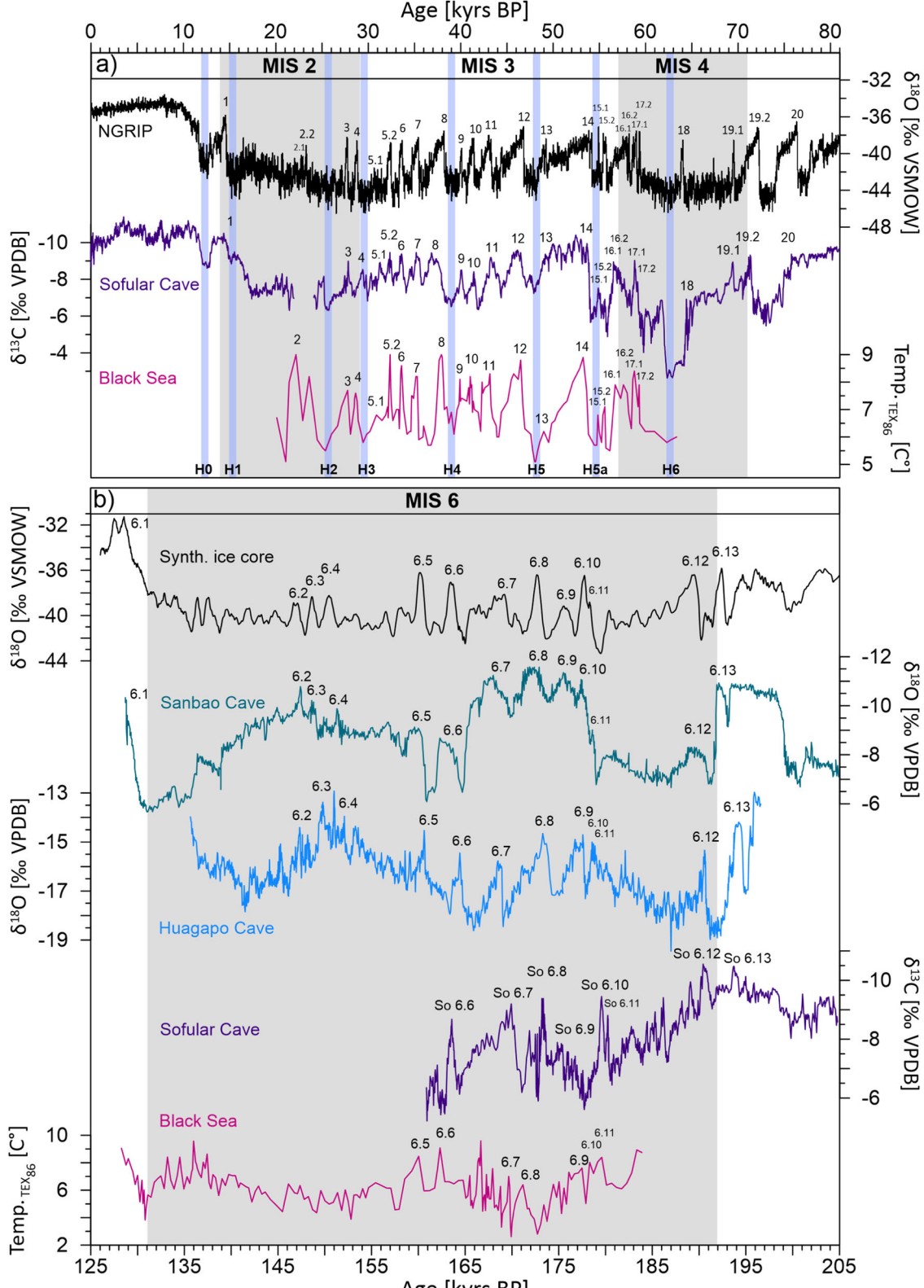

**Fig. 3 | Comparison of climate variability during the last and penultimate glacial period. a** Records showing Dansgaard-Oeschger (D-O) climate variability of the last glacial period. From top to bottom: δ[18]O of the Greenland ice core[3], δ[13]C stacked record from Sofular Cave, and a sea surface temperature (SST) record from the Black Sea[33]. Blue bars mark Heinrich events. Numbers above the records denote D-O events. **b** Records showing D-O climate variability of the penultimate glacial period. From top to bottom: δ[18]O of the synthetic ice core[22], δ[18]O of Sanbao Cave[16], δ[18]O of Huagapo Cave[17], δ[13]C stacked record from Sofular Cave, and a Black Sea SST record[43]. Numbers above the records denote D-O events. Source data are provided in the Source Data file.

and particularly during MIS 6. This effect is also apparent in Sofular stalagmites So-1 and So-2 δ¹⁸O profiles which show a suppressed Bølling-Allerød and Younger Dryas isotopic pattern[31] (Fig. 2).

In contrast, D-O cycles are more clearly visible in the Sofular δ¹³C isotope profiles indicating a high sensitivity of δ¹³C to D-O climate variability. Millennial-scale δ¹³C fluctuations in Sofular speleothems are similar between the last and penultimate glacial ranging from −5‰ to −10‰. Though So-4 and So-57 δ¹³C profiles show an almost identical isotopic pattern, δ¹³C values of stalagmite So-57 are on average 1.3‰ more positive. This isotopic offset is most likely related to sample-specific effects caused by lower drip rates and stronger fractionation due to $CO_2$ degassing. In order to reduce sample-specific effects, age uncertainties and to increase the signal-to-noise ratio, we developed a stacked δ¹³C record of MIS 1-4 and MIS 6 using the program called intra-site correlation age modeling (*iscam*)[32]. *Iscam* was designed to synchronize overlapping time series from the same site within their age uncertainties. D-O cycles in the Sofular δ¹³C stacked record are characterized by high amplitude shifts of up to 6‰ (Fig. 3a). We suggest that these shifts are primarily caused by temperature- and moisture-related changes in vegetation and soil microbial activity, with higher vegetation density and soil respiration rates associated with warmer and wetter GIs (Fig. 3a). Our assumption is based on the following lines of evidence. Firstly, a close coupling between climate and soil respiration rates is directly supported by paired δ¹³C and ¹⁴C measurements in stalagmite So-1[36]. The paired measurements reveal a rapid response of soil respiration rates and microbial activity across the Bølling-Allerød and Younger Dryas in stalagmite So-1, with enhanced decomposition of soil organic matter (SOM) under warmer and wetter climatic conditions[36].

Secondly, soil respiration rate is closely linked to temperature and soil moisture conditions on a range of timescales[37,38], a relationship that is supported by modeling experiments[39] and speleothem records from western Europe[40-42]. Thirdly, the Sofular δ¹³C stacked records show an inverse correlation ($r^2$) of -0.7 (MIS 2-4) and -0.5 (MIS 6) with sea surface temperature (SST) records from the Black Sea (gravity core GC25-1 & MSM33)[33,43] (Fig. 4, Supplementary Figs. 3 and 4). Such a close coupling between air temperature above Sofular Cave and Black Sea SSTs can be expected as the cave site is only ~10 km inland. Therefore, positive SST shifts of up to 4 °C associated with GIs[33] (Fig. 3) would lead to a marked increase in temperature and precipitation above Sofular Cave, thereby promoting a change in the type and density of vegetation[44] and an increase in soil respiration rates due to higher soil microbial activity and faster decomposition of SOM[36]. In contrast, cold-dry climatic conditions would affect vegetation density and type

and reduce $CO_2$ production due to lower microbial activity, and sparser vegetation is expected to result in lower soil $pCO_2$, lower contribution of biogenic carbon, and, consequently, higher stalagmite δ¹³C values.

The Sofular stack (Fig. 3a) exhibits the most positive δ¹³C values of around −3‰ at ~62 kyrs BP, concomitant with a major advance of the Eurasian Ice Sheet (EIS)[45,46] and Heinrich event 6. In the Black Sea region, this interval was dominated by steppe pollen taxa[44] and higher ice-rafted debris in Black Sea sediments[47]. Heinrich events H5 – H0 are also well expressed in the Sofular δ¹³C profile, and characterized by marked SST decreases of the Black Sea (Fig. 3a) and reduced precipitation in the Eastern Mediterranean[33,47,48]. Climate simulations suggest colder and drier conditions induced by a cyclonic atmospheric circulation anomaly modulating the eastward advection of cold air over Eurasia caused by a weakening of the AMOC during Heinrich events and GSs[49-52].

## D-O cycles during MIS 6

During MIS 6, a D-O like climate variability is apparent in the δ¹³C records of stalagmites So-4 and So-57. Similar to MIS 2-4, D-O cycles during MIS 6 are characterized by abrupt negative shifts of up to 4‰ in δ¹³C (Fig. 3b) in response to warmer and more humid climatic conditions and higher soil respiration rates, denser vegetation and higher proportions of C3 plant vegetation[36]. Phases of increasing temperature and higher effective moisture are broadly consistent with pollen evidence (*dec. Quercus, Betula, Pinus*) from Lake Van[53] (Supplementary Fig. 2a).

D-O cycles in the Sofular stack mirror those of the synthetic Greenland ice core δ¹⁸O record[22] (Fig. 3b), which was tuned to the absolutely-dated Sanbao Cave record[11,54]. Radiometrically dated records showing a clear D-O like pattern are currently very rare and their temporal resolution and/or chronological precision are not sufficient to identify D-O cycles[26,27,55,56]. Thus, records that can be used for a detailed comparison to Sofular are rather scarce. Suitable time series are speleothem δ¹⁸O records from Sanbao Cave[16] in China and Huagapo Cave[17] in Peru (Fig. 3b). Taking advantage of the precise and absolute chronologies of these stalagmite records, including our Sofular stack, we can gain further information on the pacing and timing of D-O like events during MIS 6. The comparison between the records show age offsets of several centuries for D-O 6.6, 6.8, 6.9 and 6.12 (Supplementary Fig. 5). Higher age offsets between the records of almost two millennia are observed for D-O 6.7, 6.10, 6.11, 6.13 and 6.14, likely caused by a combination of uncertainties related to the age models and uncertain assignment of D-O events.

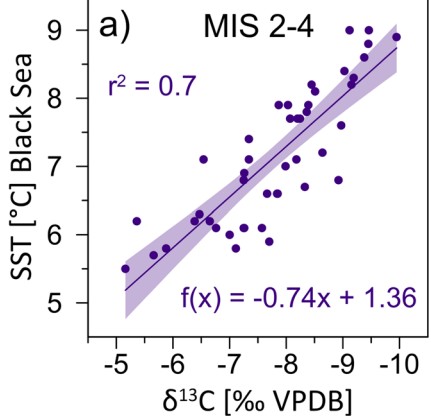
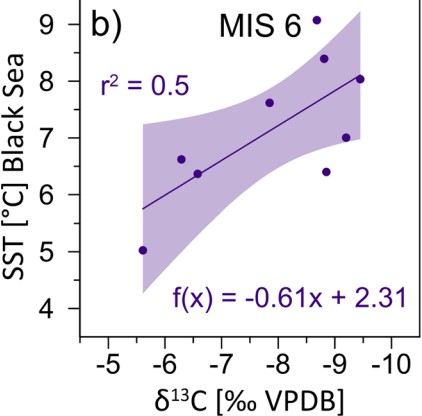

**Fig. 4 | Correlation between Sofular δ¹³C and sea surface temperature (SST) from the Black Sea.** The scatterplots show Sofular δ¹³C values of Dansgaard-Oeschger (D-O) onsets, midpoints and peaks vs sea surface temperatures (SST) from the Black Sea[33,43] during **a** the Marine Isotope Stage (MIS) 2-4 ($n = 45$ and $p \ll 0.01$) and **b** Marine Isotope Stage (MIS) 6 ($n = 9$ and p - 0.04). Shaded areas indicate the 95% confidence interval.

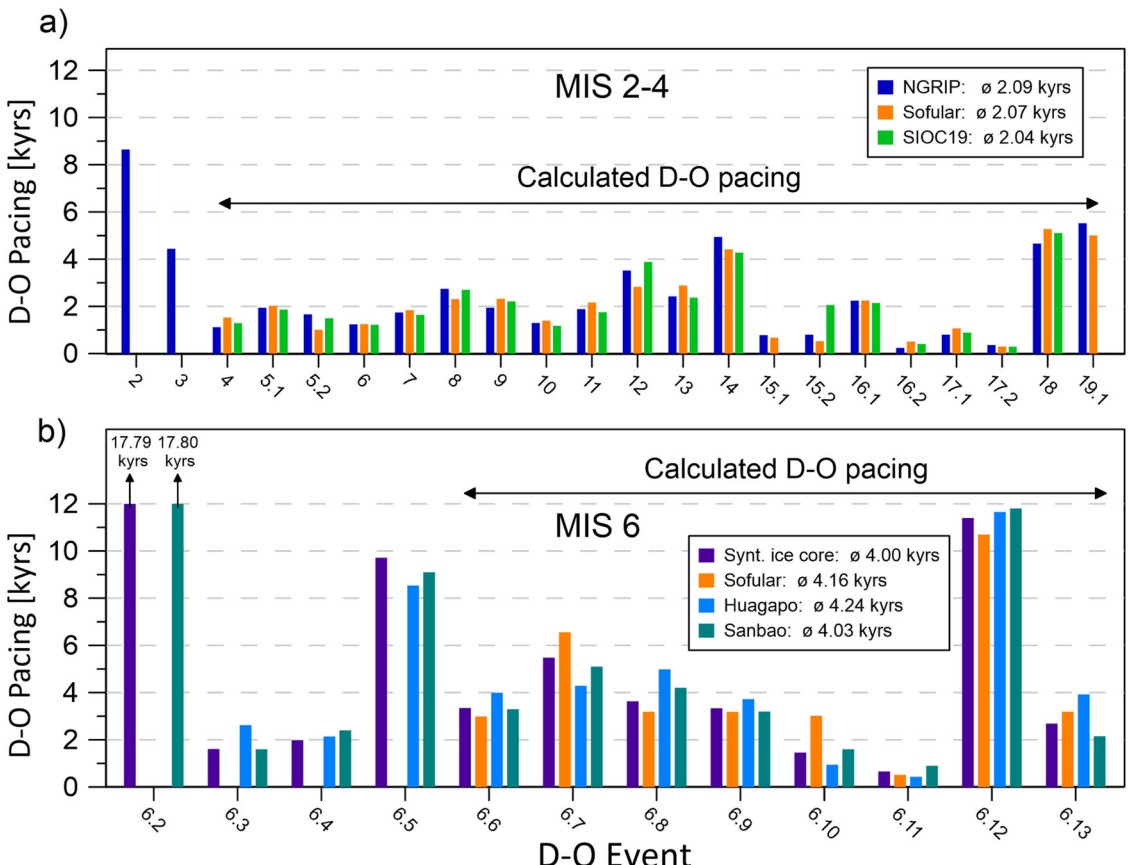

**Fig. 5 | Pacing of Dansgaard-Oeschger (D-O) cycles in Sofular and comparison of stalagmite and ice core chronologies.** Comparison of D-O pacing between: **a** D-O cycles during the last glacial period, recorded in the Greenland ice core NGRIP (data from Rasmussen et al.[3]), the Sofular stacked record and a Speleothem Interstadial Onset Compilation (SIOC19) data set (data from Corrick et al.[15]) and **b** D-O cycles of the penultimate glacial period, recorded in the synthetic ice core record[22], the Sofular stacked record, the Huagapo Cave speleothem record[17] and the Sanbao Cave speleothem record[16]. Numbers within the legends refer to the average pacings of D-O events which are taken into account by black arrows.

For the time interval between 200 and 160 kyrs BP, the average pacing of D-O cycles is ~4.16 kyrs in the Sofular stack (Fig. 5b, Supplementary Fig. 6), very similar to the pacing of ~4.24 kyrs and ~4.03 kyrs in the Huagapo and Sanbao speleothem records, respectively. All speleothem-based estimates for the D-O pacing are in agreement with a pacing of ~4.00 kyrs in the synthetic Greenland ice core. In contrast, the average D-O pacing recorded by the Sofular stack during MIS 2-4 is ~2.07 kyrs (Fig. 5a, Supplementary Fig. 7), and therefore only half as long compared to MIS 6. The Speleothem Interstadial Onset Compilation data set (SIOC19) and Greenland ice cores show an almost identical D-O pacing of ~2.04 kyrs and ~2.09 kyrs respectively (Fig. 5a). Thus, the absolutely dated Sofular stack record presented here provides additional evidence for a longer pacing of D-O cycles during MIS 6. Since D-O events are closely linked to changes in the strength of AMOC, which in turn is dependent on the rate of North Atlantic Deep Water (NADW) formation and sea-ice extent[52], the longer pacing of D-O cycles is most likely directly related to a different AMOC setting during MIS 6. Benthic foraminifera isotope data from the Portuguese margin suggest a shallower and weaker AMOC cell during MIS 6 than during MIS 3[13]. This is in agreement with longer bipolar seesaw events[57], greater sea ice cover, and lower amplitude SST variability in the South Atlantic[58]. Furthermore, iceberg discharge into the North Atlantic was muted during the first half of MIS 6 and 'classic' Heinrich events are missing[13,57]. Overall, there is strong evidence for a weaker AMOC during the penultimate glaciation. AMOC weakening leads to a considerable cooling in the North Atlantic realm and sea-ice advance over the Labrador and Nordic Seas, which in turn increases surface albedo and reduces heat loss from the ocean to the atmosphere[52]. As a result of these different mean climatic and glacial boundary conditions during MIS 6, thresholds for triggering D-O cycles were most likely higher and led to less frequent and longer pacing of D-O cycles. However, additional and more detailed reconstructions of AMOC intensity during MIS 6 are urgently needed to prove our hypothesis.

In conclusion, the stacked Sofular $\delta^{13}$C record shows D-O climate variability for the last and penultimate glacial in great detail and thereby provides additional evidence for D-O variability in the synthetic Greenland ice core record. The comparison of absolutely dated speleothem records covering MIS 6, including the Sofular stack, reveals an agreement in both the timing and pacing of D-O cycles. Thus, there is mounting evidence for a significantly longer pacing of D-O cycles during MIS 6 compared to MIS 2-4. This could be related to a generally weaker and shallower AMOC and reduced northward heat transport and significant cooling of the North Atlantic realm, thereby inducing a lower pacing of D-O events during MIS 6.

## Methods
### Stalagmite samples
A total of six stalagmites from Sofular Cave (So-1, So-2, So-4, So-6, So-13, and So-57) with heights ranging between 0.8 and 1.7 meters were analyzed. So-1, So-2, and parts of So-4 and So-6 have been investigated in previous studies[30,31], while this study provides complementary $\delta^{13}$C profiles on So-4 and So-6, and previously unpublished data of stalagmites So-13 and So-57.

## $^{230}$Th dating and age model development

U-series dating ($^{230}$Th) was performed on a multi-collector inductively coupled plasma mass spectrometer (MC-ICP-MS, Thermo-Finnigan-Neptune) at the Department of Geology and Geophysics, University of Minnesota and Institute of Global Environmental Change, Xi'an Jiaotong University (Supplementary Dataset 1). Further $^{230}$Th dating on stalagmites So-1 and So-2 was done on a Nu Instruments® MC-ICP-MS at the Geological Institute, University of Bern (Supplementary Dataset 2). Detailed information on analytical procedures is provided in Supplementary Texts 1 and 2 accompanying this article. Age models of all stalagmites were constructed using the *StalAge* algorithm[59] and the Sofular stacked record was developed using the *iscam* algorithm[32].

## Stable isotope analysis

Stable isotope analyzes were performed on a Finnigan Delta V and Delta V Plus IRMS equipped with an automated carbonate preparation system (Gas Bench-II) at the Institute of Geological Sciences, University of Bern, and Department of Environmental Sciences, University of Basel. The precision of $\delta^{13}$C and $\delta^{18}$O measurements is 0.06% and 0.07% ($1\sigma$-error) respectively.

## Data availability

Source data for Fig. 2 and Fig. 3 are referenced in the Source data provided with this paper and are available online on the NOAA paleoclimate database (https://www.ncei.noaa.gov/access/paleo-search). Source data are provided with this paper.

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

## Acknowledgements
Support of the Swiss National Science Foundation (PP002-110554/1 to D.F.), the University of Basel (to D.F.), the US National Science Foundation (NSF 2202913 to R.L.E), the National Natural Science Foundation of China (NSFC 41888101 and NSFC 42150710534 to H.C.), the Gary Comer Science and Education Foundation (CP41 to R.L.E.) and Istanbul Technical University (ITU-BAP-332491 to O.T.) facilitated this work. We would like to thank The Zonguldak Coal Geopark Administration for the permissions and helps with fieldwork.

## Author contributions
D.F. and O.T. initiated the project. K.K. helped to organize and carry out the fieldwork together with D.F. and F.H. D.F. and F.H. performed the stable isotope analysis. H.C., R.L.E., and F.H. conducted the uranium-series analysis. F.H. and D.F. wrote the paper.

## Competing interests
The authors declare no competing interests.
