## [Peer Review File · Nature Communications]

Dansgaard-Oeschger cycles of the penultimate and last glacial period recorded in stalagmites from TürkiyeREVIEWER COMMENTS

Reviewer #1 (Remarks to the Author):

Review of Held et al.

Held et al., present $\delta^{18}O$ and $\delta^{13}C$ records from Sofular cave speleothems. The speleothems cover 0-80 ka, 120-135ka and 160-220ka. The speleothems resolve the DO variability of the last deglaciation, last glacial period as well as MIS6. In addition, using a SST record from the Black Sea, the authors derive a transfer function to be able to estimate changes in air temperature above the cave. On top of evidencing D-O variability during MIS6, they suggest that the D-O pacing was twice as long during MIS6 than MIS2-4. They attribute the main differences in D-O variability between MIS6 and MIS2-4 to the different extent of the Eurasian ice-sheet (EIS).

The MIS6 record presented is very interesting and will strongly benefit the community. While I agree with the last sentence of the manuscript that the newly presented Sofular cave record could become one of the key paleoclimate timeseries for DO study given its resolution and coverage, additional work needs to be done on the manuscript before publication. Please find some comments/suggestions below.

1) The authors attribute the different expression of DO variability during MIS6 to the larger extent of the Eurasian ice-sheet. First, MIS6 is taken as one period, whereas during its 60 kyrs conditions most likely varied significantly and probably so did the EIS. The authors should thus precisely present the evidence for a larger EIS and more clearly provide information on its possible growth and decay during MIS6. I can see that an EIS volume curve is included in Fig. S4 (the reference to the EIS study is missing), but for the period of availability of the Sofular record (ie 160-200ka), I am not sure the EIS is larger than during MIS3.

L. 193-196: to which "atmospheric teleconnections" are the authors referring to? This sentence needs to be more precise.

L. 254-256: While the speleothems provide temperature estimates, the study cannot attribute the differences between MIS2-4 and MIS6 to the EIS.

On a similar topic, the authors mention in the abstract and L. 159-161 that the different D-O pacing at MIS2-4 and MIS6 is due to ice-sheet dynamics, but there is no information in the manuscript on how this conclusion is reached.

L. 231-232: I do not understand how/why it is suggested that it causes longer transition time to interstadial.

L. 253-254: How can it be concluded that the longer DO in MIS6 are due to a feedback between AMOC intensity and ice-sheet dynamics?

2) A lot of climate modelling work has been done to understand D-O variability as well as the climatic impact of changes in the Atlantic Meridional Overturning Circulation. The authors should familiarize themselves with the results of these studies to help with the understanding of their records and put their record in the context of the dynamical conclusions that have been drawn by these studies. A recent review on the topic (Menviel et al., 2020, Nature Reviews) could be a good entry point. The study also includes extensive references on appropriate climate modelling studies such as Stouffer et al. 2006, Kageyama et al., 2013...

For example, L. 115-118: previous studies (including Fleitmann et al. 2009, but also Menviel et al., 2014 Climate of the Past, Stockhecke et al., 2016 QSR and 2021 have shown the link between AMOC changes, DO variability and climate change in Turkey).

L. 168-169 could be rephrased as modelling studies have shown that an AMOC shutdown reduces the meridional oceanic heat transport to the North Atlantic, thus contributing to an increase in sea-ice

cover in the North Atlantic.

The paragraph L. 225 thus needs to be revisited. While the temperature changes mentioned are correct, they do not simply depend on latitude, but on the processes at play. References to additional climate model simulations would strengthen the study.

3) Transfer function

If I understand correctly and based on Fig. 5, the authors derive different transfer functions for MIS2-4 and MIS6. What is the rationale behind this? The use of different transfer functions makes me doubt your result, i.e. the differences in temperature estimates between MIS2-4 and MIS6. I think you should i) re-assess whether the use of 2 different transfer functions is appropriate, ii) present an uncertainty band with an estimate based on the same transfer functions.

4) Paragraph starting L. 170: I find this paragraph on DO rebound events a bit confusing. In addition, I do not understand how the authors can conclude that longer DO cycles could favour rebound events (L. 172-173).

Minor points:

The text is sometimes hard to follow due to language issues, for example L. 95-96, I don't think the use of "whereas" is appropriate here.

L. 161-162: Correlation is a strong word here

L. 218-222: I find these sentences hard to understand. Please break them down and improve the logical flow.

L. 232: The transition with "However" lacks a logical flow.

Reviewer #2 (Remarks to the Author):

Review of Dansgaard-Oeschger cycles of the penultimate and last glacial period recorded in stalagmites from Turkey by Held et al.

This manuscript presents new speleothem records that purport to show DO-type variability across the sample region during the penultimate glacial period (MIS 6). Compared with results from the same cave covering MIS 3, the new results are less continuous and the correspondence between carbon and oxygen isotopes observed during MIS 3 breaks down completely. The new results are a useful addition to the growing collection of records covering MIS 6 and provide a useful source of absolute age constraint (which is rare). However, the implications and conclusions drawn from the dataset are not well developed and thus the potential impact of the study seems to be below that which could be possible.

In particular, the main conclusions seem to concern the duration of DO cycles (or events) and their magnitude, in terms of temperature change. The first of these could be important for improving our understanding of abrupt climate variability under different boundary conditions but the discussion is poorly developed and does not represent a step forward in understanding. The second conclusion does not seem to hold water, given that other temperature records from the same interval do not appear to show increased temperature variability. This point is not addressed in the study and really needs to be, before any useful conclusions can be drawn about regional differences.

Below I outline my comments and questions in order of their appearance.

Comments and questions:

Line 23 "there are almost no proxy records which provide unambiguous evidence for their existence during preceding glacial periods" – This is not the case. Some of the papers you cite later provide such evidence. Here is a short list of examples: Martrat et al., 2004; 2007; Barker et al., 2015, Cortina et al., 2015; Cheng et al., 2016

Line 88 "The new Sofular $\delta^{18}\text{O}$ and $\delta^{13}\text{C}$ records show characteristic D-O cycles, which are, even on multidecadal timescales, remarkably similar to those recorded in Greenland ice cores" – Considering the number of proxy records that look like Greenland $\delta^{18}\text{O}$ I would not say that this dataset stands out as being particularly remarkable. At the least you should quantify how good the agreement is, and more importantly, you need to describe those parts of the record that differ from the variability recorded in Greenland. For example, the amplitude of individual events is much smaller relative to the baseline difference between (some) stadial events in the Sofular record compared to that seen at Greenland. E.g. compare stadial values before and after DO8, or during H events 4, 5, 5a and 6, which contrast with the relative constancy of the Greenland record for the same events.

Same point on line 130 "Considering the very high coincidence of abrupt climate events between Greenland ice cores and Sofular stalagmites...". Use of the term 'high confidence' demands statistical quantification.

Line 104 "Such a climate amelioration would not only promote a change in the type and density of vegetation but also increase soil respiration rates due to higher soil microbial activity and faster decomposition of SOM" - How fast would this change occur? Does this limit the rapidity of changes that can be recorded?

Line 111 "The Sofular record exhibits the most positive $\delta^{13}\text{C}$ values of around -3‰ at ~62 ka BP, concomitant with the maximum extent of ice sheets in Eurasia and Heinrich event 6 (Fig. 2)." - This is confusing and conflicts with the evidence in Fig. 2 and the cited papers. Fig 2 shows maps for the LGM and MIS 6, not H6 or MIS 4 (which was smaller in extent than the LGM, according to the cited paper.

Line 115 "Since Heinrich events (HE) were most likely restricted to the midlatitude North Atlantic, evidence...". This is not true, the effects of H events were certainly not restricted to the mid northern latitudes but are seen in records around the Earth. For example, Brazilian speleothems show growth periods associated with H events (Wang et al., 2004), surface ocean frontal shifts in the SE Atlantic were more accentuated for H events than non-H events (Barker and Diz, 2014), Chinese speleothems show more pronounced variations associated with H events (Wang et al., 2001). H events are also more pronounced than non-H events in the deep ocean e.g. the deep South Atlantic (Gottschalk et al., 2015) and the NE Atlantic (Henry et al., 2016).

Line 136 "During MIS 6, a D-O like climate variability is most clearly visible in the $\delta^{13}\text{C}$ records" - But why does $\delta^{18}\text{O}$ not show these events when it does so clearly during MIS 3? This demands some explanation.

Line 137 "D-O cycles during MIS 6 are characterized by abrupt negative shifts of up to 4‰ (Fig. 3) in response to warmer and more humid climatic conditions and higher soil respiration rates, denser vegetation and higher proportions of C3 plant vegetation." – what is the evidence for this?

Line 157 "The average pacing of D-O cycles is almost twice as long (3.83 ka) during MIS 6 compared to MIS 2-4 (2.07 ka) (Fig. 4)." – I have several concerns about this statement and therefore the arguments that build on it. Firstly, from Fig. 4 it looks like you are describing the duration of individual DO events rather than the time between events (which is implied by the term 'pacing') – the distinction is very important for any discussion of the underlying causes. You repeatedly use the term 'DO cycles' (which would include both stadial and interstadial event) but if you are actually referring to

the duration of DO events, this needs to be made clear. Secondly, I could not find any description of how the duration was measured for each dataset. This needs to be shown in detail to allow the reader to assess how robust the finding is. Thirdly (and this is related the second point), looking at Figs 2 and 3, it is not at all obvious that your calculated durations are correct. For example in Fig 4, DO6.6 lasts for something like 3-5kyr but most of the records in Fig 3 show it to be 2-3kyr.

Line 161 - The following discussion requires a lot of development before it might represent a step forward in understanding. At the moment it is just a collection of vague possibilities, with no thread, or coherent argument.

Line 180 "...the Huagapo $\delta^{18}O$ record reveals a pronounced asymmetric D-O pattern during MIS 6 suggesting a significant influence of different regional boundary conditions on D-O events." - Or it could just reflect differences in the recording of DO events by different archives and proxies. This is a crucial question.

Line 187 Please stipulate which data presented here were published previously in the Badertscher 2011 paper.

Line 207 and below - This exercise makes very little sense to me. You are calibrating to a very low resolution record and you get a completely different calibration for MIS 6 vs MIS 2 (which should tell you something is wrong). Also, you omit to mention the numerous temperature (and equivalent) reconstructions from MIS 6 which are nicely summarized in the Wegwerth paper you cite for the Black Sea temperature record. None of these other reconstructions suggest a larger amplitude of temperature change associated with DOs of MIS 6. So are you challenging all of these records or are you suggesting that the larger magnitude of DO warming during MIS 6 was limited to a region around your sample site?

Minor comments

Line 38 "a very tied atmospheric teleconnection" - poor wording

Line 81 "The Sofular record covers the last 80,000 years and the period between 130,000 and 200,000 years before present almost continuously. Hiatuses exist between 21.6 - 24.1 ka BP and 134 - 160 ka BP (Figs. 2 and 3)." - A gap of 26kyr within a 70kyr interval is hardly 'almost continuous'.

References

- Barker, S., Chen, J., Gong, X., Jonkers, L., Knorr, G., & Thornalley, D. (2015). Icebergs not the trigger for North Atlantic cold events. *Nature*, 520(7547), 333-336.
- Barker, S., & Diz, P. (2014). Timing of the descent into the last ice age determined by the bipolar seesaw. *Paleoceanography*, 29(6), 489-507.
- Cheng, H., Edwards, R. L., Sinha, A., Spötl, C., Yi, L., Chen, S., Kelly, M., Kathayat, G., Wang, X., & Li, X. (2016). The Asian monsoon over the past 640,000 years and ice age terminations. *Nature*, 534(7609), 640-646.
- Cortina, A., Sierro, F. J., Flores, J. A., Martrat, B., & Grimalt, J. O. (2015). The response of SST to insolation and ice sheet variability from MIS 3 to MIS 11 in the northwestern Mediterranean Sea (Gulf of Lions). *Geophysical Research Letters*, 42(23), 10,366-310,374.
- Martrat, B., Grimalt, J. O., Lopez-Martinez, C., Cacho, I., Sierro, F. J., Flores, J. A., Zahn, R., Canals, M., Curtis, J. H., & Hodell, D. A. (2004). Abrupt temperature changes in the Western Mediterranean over the past 250,000 years. *Science*, 306(5702), 1762-1765.
- Martrat, B., Grimalt, J. O., Shackleton, N. J., de Abreu, L., Hutterli, M. A., & Stocker, T. F. (2007). Four climate cycles of recurring deep and surface water destabilizations on the Iberian margin. *Science*, 317(5837), 502-507.
- Wang, X. F., Auler, A. S., Edwards, R. L., Cheng, H., Cristalli, P. S., Smart, P. L., Richards, D. A., &

Shen, C. C. (2004). Wet periods in northeastern Brazil over the past 210 kyr linked to distant climate anomalies. *Nature*, 432(7018), 740-743.

Wang, Y. J., Cheng, H., Edwards, R. L., An, Z. S., Wu, J. Y., Shen, C. C., & Dorale, J. A. (2001). A high-resolution absolute-dated Late Pleistocene monsoon record from Hulu Cave, China. *Science*, 294(5550), 2345-2348.

Reviewer #3 (Remarks to the Author):

Held and co-authors present high-resolution and, as for these archives possible, extremely well dated speleothem isotope data from NW-Anatolia reaching back into the penultimate glacial period, to provide evidence and discuss Dansgaard-Oeschger-type climate changes during MIS 6 in comparison to the well-known D-O variability of the last glacial. Focus is mainly on the speleothem carbon isotope signal fluctuating in function of soil pCO₂, which in turn depends on temperature and moisture availability, both affecting vegetation and soil microbial activity. The authors observe an immediate response in the δ¹³C signal to the North Atlantic climate cycles, even recording in detail the precursor-type events. MIS 6 variations are compared to other stalagmite data and the synthetic Greenland ice core record derived from Antarctic ice cores. For some of the identified D-O events, age offsets are small, for others the assignments are ambiguous and age offsets are relatively large. Latter is particularly the case, where the offset in the isotope signals between two different stalagmites of the same cave (So-4 and So-57) is very large. The authors suggest that D-O climate variability and duration as recorded in Anatolia is strongly linked to insolation changes and the size of the European Ice Sheet, by affecting the teleconnection strengths to the North Atlantic system. In a last part of the manuscript, the δ¹³C signal is calibrated against Black Sea surface temperatures to discuss temperature variability at the cave site.

Since such detailed D-O observations for MIS 6 are rather sparse, this contribution is highly relevant for the comprehensive understanding of the dynamics of rapid climate change and I would very much like to see this work published in *Nature communications* after moderate revisions.

Besides some following general remarks, please find some detailed comments below.

First, being aware of the previous studies on the Sofular speleothems, I would ask the authors to more precisely state which data are original to this study and which data were used from previously published work. In particular, MIS 2-4 data and corresponding discussion is reiterated in several places and could perhaps be shortened.

Second, the authors convincingly use the δ¹³C as the better and more immediate recorder of environmental changes above the cave. But this strength is also a weakness at the same time, when it comes to reproducibility, absolute values and amplitudes. The authors indeed discuss the complexity of the δ¹³C signal formation, but at the end they calculate an average signal from two very different records, and, even more important, they use this record to linearly calibrate and extract temperature information, knowing that distinct hydroclimatic changes and shifts in the vegetation contributed significantly to the signal formation. Detailed reconstructions of the MIS 2-4 Anatolian vegetation dynamics, e.g., share the overall D-O pattern, but with significant differences as well (Shumilovskikh et al. 2014, *Climate of the Past*). Therefore, absolute temperatures as well as the amplitude (I would expect higher amplitudes on the continent and at this elevation) of the changes may differ significantly from the proposed temperature record, which is linked straight forward to Black Sea surface temperatures, without considering differences in elevation (ca. 500 m vs sea level) and a certain continentality and seasonality (since Black Sea temperatures are assumed to represent annual

averages and soil activity is definitively biased towards the warm season). Although I find the idea to extract temperatures from the stalagmite $\delta^{13}C$ a very interesting exercise, there are too many uncertainties and unknowns involved and I recommend to rather stay with the original $\delta^{13}C$. Unless one were to take the certainly promising approach of extracting organic biomarker (GDGCs) from stalagmites to reconstruct soil/atmospheric temperatures above the cave (e.g., Zang et al. 2023, DOI 10.3389/fevo.2023.1117599).

The authors discuss the MIS 6 $\delta^{18}O$ data noting that there is no asymmetry (as perhaps expected) in the D-O cycles and that large changes in the Black Sea hydrology (e.g., marine and melt water inflows) matter as well. Reference to two Black Sea studies could strengthen the discussion here, since they showed (1) that the glacial Black Sea $\delta^{18}O$ source signal has undergone a strong modification compared to the Greenland ice cores, due to the variable mixing time of the large Black Sea reservoir smoothing and delaying the signal - a one to one pattern cannot be expected (Wegwerth et al. 2019, *communications earth and environment*), and (2) that three large meltwater pulses arrived in the Black Sea during the penultimate glacial.

Comments:

L 22: change to "(AMOC) and solar and volcanic forcing.

L 63-67, 79-85: Please make clear, which part of the data are original to this paper.

L 102-109: Evidence for vegetation changes come from, e.g. Shumilovkikh et al., add ref

L 115-117: This link has been discussed widely by Wegwerth et al. 2015 and 2016, add refs

L 120-121: Please indicate more precisely precursor-type events in Fig. 2

Suppl. Fig. 1: no element ratio but only the concentrations of Mg and Sr is shown. As it is now, the statement is not supported.

L 124-126 and Suppl. Fig. 1b: this figure should contain a direct and detailed comparison between the Sofular and NGRIP isotope data to support the statement

L 141-145: Offset between So-4 and So-57 is discussed, however the stacking approach is rather questionable, since the patterns and amplitudes are significantly different. For this discussion, a composite is not needed.

L 139-140: Include pollen data von Lake Van into the graph

L 148: Statement is misleading. Of course one can compare various records to the Sofular data even if they are of lower resolution and weaker in stratigraphy. In that case, one has to consider the respective limitations.

Fig. 4 caption: change to "...synthetic ice core.

L 161-162: There are many D-O events not fitting into this picture. Please discuss.

L 176-178: See general comment on the $\delta^{18}O$ record

L 183-186: In fact, three periods of melt water input into the Black Sea are hypothesized for MIS 6

L 225-236: Regional patterns in D-O associated temperature changes have been summarized and thoroughly discussed in Wegwerth et al. 2016.

Responses to Reviewers

We would like to thank all reviewers for their thoughtful and constructive comments, most of which helped to significantly improve the quality of our manuscript. Please find our responses (in black) to the reviewer comments (in blue).

Based on their comments and suggestions, we did a major revision of our manuscript, and also added and changed most figures. We also adjusted the general structure to address the comments of the reviewers. In particular, we agree with the concerns of the reviewers that the temperature reconstruction is subject to major uncertainties and limitations. We therefore removed this part of the manuscript as it was too controversial. To improve and sharpen the content of our manuscript, we acquired new additional data by adding another stalagmite So-13 and by improving the chronology of So-57 with more ^{230}Th ages.

Reviewer: 1

1) The authors attribute the different expression of DO variability during MIS6 to the larger extent of the Eurasian ice-sheet. First, MIS6 is taken as one period, whereas during its 60 kyrs conditions most likely varied significantly and probably so did the EIS. The authors should thus precisely present the evidence for a larger EIS and more clearly provide information on its possible growth and decay during MIS6. I can see that an EIS volume curve is included in Fig. S4 (the reference to the EIS study is missing), but for the period of availability of the Sofular record (ie 160-200ka), I am not sure the EIS is larger than during MIS3.

We apologize that we didn't make it clearer and we agree with reviewer 1 that the greater extent of the EIS is not responsible for the longer pacing of D-O events during MIS 6. In order to clarify this, we expanded the discussion on the potential causes for longer D-O pacings in terms of changes in the AMOC strength and bipolar sea saw mechanism related to the interhemispheric coupling.

L. 193-196: to which "atmospheric teleconnections" are the authors referring to? This sentence needs to be more precise.

We are referring to the atmospheric teleconnection between the North Atlantic and Eastern Mediterranean. We changed the sentence accordingly.

L. 254-256: While the speleothems provide temperature estimates, the study cannot attribute the differences between MIS2-4 and MIS6 to the EIS.

This is now obsolete as we discarded our controversial temperature reconstruction.

On a similar topic, the authors mention in the abstract and L. 159-161 that the different D-O pacing at MIS2-4 and MIS6 is due to ice-sheet dynamics, but there is no information in the manuscript on how this conclusion is reached.

We would like to thank reviewer 1 for this thoughtful comment. We have refined the discussion on the relationship between longer D-O pacings and changes in the Atlantic Meridional Ocean Circulation (AMOC). We included additional references (Menviel et al., 2020; Shin et al., 2020; Margari et al., 2010; Gottschalk et al., 2020) to highlight and discuss important components contributing to differences in the AMOC state during MIS 2-4 and MIS 6.

L. 231-232: I do not understand how/why it is suggested that it causes longer transition time to interstadial.

We do not discuss this matter in the revised version of the manuscript, also because we removed the quantitative temperature reconstruction.

L. 253-254: How can it be concluded that the longer DO in MI6 are due to a feedback between AMOC intensity and ice-sheet dynamics?

See comment above. In the revised version we do not discuss the duration of D-O events but rather focus on their pacing.

2) A lot of climate modelling work has been done to understand D-O variability as well as the climatic impact of changes in the Atlantic Meridional Overturning Circulation. The authors should familiarize themselves with the results of these studies to help with the understanding of their records and put their record in the context of the dynamical conclusions that have been drawn by these studies. A recent review on the topic (Menviel et al., 2020, Nature Reviews) could be a good entry point. The study also includes extensive references on appropriate climate modelling studies such as Stouffer et al. 2006, Kageyama et al., 2013...

We thank reviewer 1 for this interesting reference and we have expanded our discussion to address the potential causes of AMOC changes on D-O climate variability.

For example, L. 115-118: previous studies (including Fleitmann et al. 2009, but also Menviel et al., 2014 Climate of the Past, Stockhecke et al., 2016 QSR and 2021 have shown the link between AMOC changes, DO variability and climate change in Turkey).

Addressed in the revised version, see also comment above.

L. 168-169 could be rephrased as modelling studies have shown that an AMOC shutdown reduces the meridional oceanic heat transport to the North Atlantic, thus contributing to an increase in sea-ice cover in the North Atlantic.

Revised.

The paragraph L. 225 thus needs to be revisited. While the temperature changes mentioned are correct, they do not simply depend on latitude, but on the processes at play. References to additional climate model simulations would strengthen the study.

Due to the reviewers' concerns about the uncertainty of the temperature dataset, we decided to exclude it from the revision and refrain from further discussion.

3) Transfer function

If I understand correctly and based on Fig. 5, the authors derive different transfer functions for MIS2-4 and MIS6. What is the rationale behind this? The use of different transfer functions makes me doubt your result, i.e. the differences in temperature estimates between MIS2-4 and MIS6. I think you should i) re-assess whether the use of 2 different transfer functions is appropriate, ii) present an uncertainty band with an estimate based on the same transfer functions.

This comment is now obsolete because we decided to remove the controversial quantitative temperature reconstruction, a transfer function is therefore not used in the revised manuscript. We only show a crossplot between Black Sea surface temperature and stalagmite $\delta^{13}\text{C}$ to highlight the strong influence of temperature on carbon isotope values.

4) Paragraph starting L. 170: I find this paragraph on DO rebound events a bit confusing. In addition, I do not understand how the authors can conclude that longer DO cycles could favour rebound events (L. 172-173).

We recognized that this discussion is not highly relevant for the core of our manuscript, as the main focus is on D-O variability during MIS 6. For this reason, we have decided to remove the discussion of precursor and rebound from the revised version.

Minor points:

The text is sometimes hard to follow due to language issues, for example L. 95-96, I don't think the use of "whereas" is appropriate here.

Corrected.

L. 161-162: Correlation is a strong word here

Due to a high uncertainty in the previously presented relationship between D-O pacings and insolation and the fact that we cannot provide a clear conclusion and further discussion on the current data basis, we decided to discuss the potential causes for longer D-O cycles from a different point of view focusing on interlinked changes in the AMOC-sea-ice-atmosphere system.

L. 218-222: I find these sentences hard to understand. Please break them down and improve the logical flow.

Obsolete, the sentence belonging to the quantitative temperature estimates is removed from the revised manuscript.

L. 232: The transition with "However" lacks a logical flow.

Removed.

Reviewer: 2

Line 23 “there are almost no proxy records which provide unambiguous evidence for their existence during preceding glacial periods” – This is not the case. Some of the papers you cite later provide such evidence. Here is a short list of examples: Martrat et al., 2004; 2007; Barker et al., 2015, Cortina et al., 2015; Cheng et al., 2016

Changed, we are now stating “...only very few terrestrial records...” to avoid any misunderstanding.

Line 88 “The new Sofular $\delta^{18}\text{O}$ and $\delta^{13}\text{C}$ records show characteristic D-O cycles, which are, even on multidecadal timescales, remarkably similar to those recorded in Greenland ice cores” – Considering the number of proxy records that look like Greenland $\delta^{18}\text{O}$ I would not say that this dataset stands out as being particularly remarkable. At the least you should quantify how good the agreement is, and more importantly, you need to describe those parts of the record that differ from the variability recorded in Greenland. For example, the amplitude of individual events is much smaller relative to the baseline difference between (some) stadial events in the Sofular record compared to that seen at Greenland. E.g. compare stadial values before and after DO8, or during H events 4, 5, 5a and 6, which contrast with the relative constancy of the Greenland record for the same events.

We have changed the wording from “remarkably” to “very” to avoid any misunderstandings.

Same point on line 130 “Considering the very high coincidence of abrupt climate events between Greenland ice cores and Sofular stalagmites...”. Use of the term ‘high confidence’ demands statistical quantification.

We are not using the term “High confidence” in our manuscript.

Line 104 “Such a climate amelioration would not only promote a change in the type and density of vegetation but also increase soil respiration rates due to higher soil microbial activity and faster decomposition of SOM” - How fast would this change occur? Does this limit the rapidity of changes that can be recorded?

The time lag between the climate amelioration and the vegetation change is relatively short for most of the observed DO events, as suggested by the rapid $\delta^{13}\text{C}$ changes observed and also by the good synchronicity between $\delta^{13}\text{C}$ and $\delta^{18}\text{O}$ records. Furthermore, pollen records from the Black Sea area (Shumilovskikh et al., 2014) and Lake Van show an almost synchronous and immediate response of vegetation within decades to D-O events. Other records from the Mediterranean (Allen et al., 1999) and marine pollen data from the Iberian Margin (Sanchez Goni et al., 2002) show very similar short time lags between climate and vegetation changes.

Line 111 “The Sofular record exhibits the most positive $\delta^{13}\text{C}$ values of around -3‰ at ~62 ka BP, concomitant with the maximum extent of ice sheets in Eurasia and Heinrich event 6 (Fig. 2).” - This is confusing and conflicts with the evidence in Fig. 2 and the cited papers. Fig 2 shows maps for the LGM and MIS 6, not H6 or MIS 4 (which was smaller in extent than the LGM, according to the cited paper.

We apologize for this confusing statement, we changed the wording to “...a major advance of the Eurasian Ice Sheet (EIS)...”.

Line 115 “Since Heinrich events (HE) were most likely restricted to the midlatitude North Atlantic, evidence...”. This is not true, the effects of H events were certainly not restricted to the mid northern latitudes but are seen in records around the Earth. For example, Brazilian speleothems show growth periods associated with H events (Wang et al., 2004), surface ocean frontal shifts in the SE Atlantic were more accentuated for H events than non-H events (Barker and Diz, 2014), Chinese speleothems

show more pronounced variations associated with H events (Wang et al., 2001). H events are also more pronounced than non-H events in the deep ocean e.g. the deep South Atlantic (Gottschalk et al., 2015) and the NE Atlantic (Henry et al., 2016).

We agree with reviewer 2 that this statement is misleading. We removed the sentence.

Line 136 “During MIS 6, a D-O like climate variability is most clearly visible in the $\delta^{13}\text{C}$ records” - But why does $\delta^{18}\text{O}$ not show these events when it does so clearly during MIS 3? This demands some explanation.

In the revised version we have expanded the discussion to include further explanations for the lack of a clear D-O pattern in the Sofular oxygen isotope records. The new paragraph states that: “...Thus, the “water vapour source effect” in combination with variable mixing times of the Black Sea³⁴ dampened D-O related fluctuations in $\delta^{18}\text{O}$ during MIS 6. This effect is also apparent in Sofular stalagmites So-1 and So-2 $\delta^{18}\text{O}$ profiles which show a suppressed Bølling-Allerød and Younger Dryas isotopic pattern²⁹ (Fig. 2).”

Line 137 “D-O cycles during MIS 6 are characterized by abrupt negative shifts of up to 4‰ (Fig. 3) in response to warmer and more humid climatic conditions and higher soil respiration rates, denser vegetation and higher proportions of C3 plant vegetation.” – what is the evidence for this?

As the same climatic and environmental factors like an increase in temperature, moisture and soil respiration were responsible for $\delta^{13}\text{C}$ fluctuations during MIS 2-4, it is therefore most likely that the same factors influenced $\delta^{13}\text{C}$ values during MIS 6. Furthermore, the close correlation between $\delta^{13}\text{C}$ and Black Sea sea surface temperatures supports our assumption. Finally, the statement is supported by the following sentence regarding the pollen record of Lake Van. We added “Phases of increasing temperature and higher effective moisture are consistent with pollen evidence (*dec. Quercus, Betula, Pinus*) from Lake Van⁵³.”

Line 157 “The average pacing of D-O cycles is almost twice as long (3.83 ka) during MIS 6 compared to MIS 2-4 (2.07 ka) (Fig. 4).” – I have several concerns about this statement and therefore the arguments that build on it. Firstly, from Fig. 4 it looks like you are describing the duration of individual DO events rather than the time between events (which is implied by the term ‘pacing’) – the distinction is very important for any discussion of the underlying causes. You repeatedly use the term ‘DO cycles’ (which would include both stadial and interstadial event) but if you are actually referring to the duration of DO events, this needs to be made clear. Secondly, I could not find any description of how the duration was measured for each dataset. This needs to be shown in detail to allow the reader to assess how robust the finding is. Thirdly (and this is related the second point), looking at Figs 2 and 3, it is not at all obvious that your calculated durations are correct. For example in Fig 4, DO6.6 lasts for something like 3-5kyr but most of the records in Fig 3 show it to be 2-3kyr.

We apologize for the incorrect representation of Figure 4 and the resulting confusion. In fact, Figure 4 shows the pacing, and we corrected the Y-axis title. A detailed illustration showing how the pacing is determined is now available in Supplementary Figure 6 and 7.

Line 161 - The following discussion requires a lot of development before it might represent a step forward in understanding. At the moment it is just a collection of vague possibilities, with no thread, or coherent argument.

We agree with this comment to a certain extent, but think that the discussion does contribute to further understanding. Of course, we cannot draw any final conclusions at this stage, as the data base

and evidence for MIS 6 is still sparse. As reviewer 2 mentioned at the introduction, absolutely dated records covering MIS 6 are currently rare.

Line 180 "...the Huagapo $\delta^{18}\text{O}$ record reveals a pronounced asymmetric D-O pattern during MIS 6 suggesting a significant influence of different regional boundary conditions on D-O events." - Or it could just reflect differences in the recording of DO events by different archives and proxies. This is a crucial question.

Obsolete, the sentence belonging to the different D-O characteristics is removed from the revised manuscript.

Line 187 Please stipulate which data presented here were published previously in the Badertscher 2011 paper.

We have modified the introduction and methods to stipulate which data were already published in previous publications. The $\delta^{13}\text{C}$ record for time interval between 50 and 80 kyr was never presented in a publication, the same is true for the MIS 6 records. We also added two new stalagmite records (So-13 and So-57). Overall, the majority of data is original and unpublished. We adjusted the text to highlight the originality of our dataset.

Line 207 and below - This exercise makes very little sense to me. You are calibrating to a very low resolution record and you get a completely different calibration for MIS 6 vs MIS 2 (which should tell you something is wrong). Also, you omit to mention the numerous temperature (and equivalent) reconstructions from MIS 6 which are nicely summarized in the Wegwerth paper you cite for the Black Sea temperature record. None of these other reconstructions suggest a larger amplitude of temperature change associated with DOs of MIS 6. So are you challenging all of these records or are you suggesting that the larger magnitude of DO warming during MIS 6 was limited to a region around your sample site?

We agree that the current data are of limited suitability for a calibration and we removed this part from the revised version. We only show a crossplot between Black Sea SST's and Sofular $\delta^{13}\text{C}$ to demonstrate that temperature (and precipitation) have an effect on soil respiration rates and vegetation above the cave.

Minor comments

Line 38 "a very tied atmospheric teleconnection" – poor wording

Modified.

Line 81 "The Sofular record covers the last 80,000 years and the period between 130,000 and 200,000 years before present almost continuously. Hiatuses exist between 21.6 - 24.1 ka BP and 134 - 160 ka BP (Figs. 2 and 3)." – A gap of 26kyr within a 70kyr interval is hardly 'almost continuous'.

Modified.

Reviewer: 3

First, being aware of the previous studies on the Sofular speleothems, I would ask the authors to more precisely state which data are original to this study and which data were used from previously published work. In particular, MIS 2-4 data and corresponding discussion is reiterated in several places and could perhaps be shortened.

We have modified the introduction and methods accordingly. Sofular $\delta^{13}\text{C}$ isotope profiles for MIS 4 and MIS 6 were never presented in a publication. In the revised version we also added a new stalagmite records (So-13) and we improved the chronology of So-57 with new absolutely dated ^{230}Th ages. We adjusted the text to highlight the originality of our dataset.

Second, the authors convincingly use the d^{13}C as the better and more immediate recorder of environmental changes above the cave. But this strength is also a weakness at the same time, when it comes to reproducibility, absolute values and amplitudes. The authors indeed discuss the complexity of the d^{13}C signal formation, but at the end they calculate an average signal from two very different records, and, even more important, they use this record to linearly calibrate and extract temperature information, knowing that distinct hydroclimatic changes and shifts in the vegetation contributed significantly to the signal formation. Detailed reconstructions of the MIS 2-4 Anatolian vegetation dynamics, e.g., share the overall D-O pattern, but with significant differences as well (Shumilovskikh et al. 2014, *Climate of the Past*). Therefore, absolute temperatures as well as the amplitude (I would expect higher amplitudes on the continent and at this elevation) of the changes may differ significantly from the proposed temperature record, which is linked straight forward to Black Sea surface temperatures, without considering differences in elevation (ca. 500 m vs sea level) and a certain continentality and seasonality (since Black Sea temperatures are assumed to represent annual averages and soil activity is definitively biased towards the warm season). Although I find the idea to extract temperatures from the stalagmite d^{13}C a very interesting exercise, there are too many uncertainties and unknowns involved and I recommend to rather stay with the original d^{13}C . Unless one were to take the certainly promising approach of extracting organic biomarker (GDGCs) from stalagmites to reconstruct soil/atmospheric temperatures above the cave (e.g., Zang et al. 2023, DOI 10.3389/fevo.2023.1117599).

Like reviewer 2, reviewer 3 questions the development of a quantitative temperature reconstruction from Sofular stalagmites. We removed this part of the manuscript in order to focus more on the nature of D-O events during MIS 6. It is, however, common practices in paleoclimatology to use multiple speleothem records to develop a stacked record. This approach helps to minimize sample-specific signals and therefore to increase the signal to noise ratio. As mentioned above, this is, for instance, frequently done for tree ring records and marine sediment records (e.g., SPECMAP and LR 04 stack).

The authors discuss the MIS 6 d^{18}O data noting that there is no asymmetry (as perhaps expected) in the D-O cycles and that large changes in the Black Sea hydrology (e.g., marine and melt water inflows) matter as well. Reference to two Black Sea studies could strengthen the discussion here, since they showed (1) that the glacial Black Sea d^{18}O source signal has undergone a strong modification compared to the Greenland ice cores, due to the variable mixing time of the large Black Sea reservoir smoothing and delaying the signal - a one to one pattern cannot be expected (Wegwerth et al. 2019, *communications earth and environment*), and (2) that three large meltwater pulses arrived in the Black Sea during the penultimate glacial.

We thank reviewer 3 for the comment and have expanded the discussion.

Comments:

L 22: change to "(AMOC) and solar and volcanic forcing.

Corrected.

L 63-67, 79-85: Please make clear, which part of the data are original to this paper.

See first comment.

L 102-109: Evidence for vegetation changes come from, e.g. Shumilovkikh et al.,. add ref

Corrected.

L 115-117: This link has been discussed widely by Wegwerth et al. 2015 and 2016, add refs

Changed.

L 120-121: Please indicate more precisely precursor-type events in Fig. 2

We recognized that the paragraph on precursor and rebound events disturbs the general structure and that this discussion is not highly relevant for the core of our manuscript. For this reason, we decided to remove the discussion of precursor and rebound from the revised version.

Suppl. Fig. 1: no element ratio but only the concentrations of Mg and Sr is shown. As it is now, the statement is not supported.

See comment above.

L 124-126 and Supp. Fig. 1b: this figure should contain a direct and detailed comparison between the Sofular and NGRIP isotope data to support the statement

See comment above.

L 141-145: Offset between So-4 and So-57 is discussed, however the stacking approach is rather questionable, since the patterns and amplitudes are significantly different. For this discussion, a composit is not needed.

The use of a stacked record is particularly advantageous for records with different amplitudes, as sample-specific effects are reduced. This improves the reliability of both records in terms of age uncertainty and increases the signal to noise ratio. As mentioned above, this is a common and desirable practice in paleoclimatology.

L 139-140: Include pollen data von Lake Van into the graph

Lake Van is mentioned in one sentence and we therefore decided to include the pollen data from Lake Van in Supplementary Figure 2.

L 148: Statement is misleading. Of course one can compare various records to the Sofular data even if they are of lower resolution and weaker in stratigraphy. In that case, one has to consider the respective limitations.

Modified.

Fig. 4 caption: change to ...synthetic ice core.

Corrected.

L 161-162: There are many D-O events not fitting into this picture. Please discuss.

We agree with reviewer 3 that the high uncertainty in the previously presented relationship between D-O pacing and solar insolation is insufficient for an informed conclusion. Since we cannot provide a clear conclusion and further discussion on the currently available data, we decided to consider the relationship of longer D-O cycles and ice dynamics from a different point of view.

L 176-178: See general comment on the d18O record

Modified.

L 183-186: In fact, three periods of melt water input into the Black Sea are hypothesized for MIS 6

Modified, information provided in Supplementary Figure 2.

L 225-236: Regional patterns in D-O associated temperature changes have been summarized and thoroughly discussed in Wegwerth et al. 2016.

Obsolete, the sentence belonging to the quantitative temperature estimates is removed from the revised manuscript.

REVIEWER COMMENTS

Reviewer #1 (Remarks to the Author):

2nd review of Held et al.

In their revised version, Held et al addressed my comments related to the link with modelling and relationship between $\delta^{13}\text{C}$ and temperature. Please find below a few minor comments that should be addressed before publication.

Abstract, L. 12-15: A weaker AMOC as the reason behind the longer DO pacing is an hypothesis that cannot be tested here. I thus suggest to carefully phrase it as such (and also L. 227-230). In addition here "background conditions" are also mentioned whereas they are not mentioned L. 227-230 (unless I misunderstood what you meant by background conditions).

L. 22: There is a lot of evidence for DO cycles being associated with AMOC changes. What caused these AMOC changes is unclear, and volcanic forcing is one of the hypothesis being put recently forward but has only limited evidence, but not solar cycles.

L. 23-24: I understand there are few terrestrial records of the penultimate glacial period which present DO variability but since they look at pollen, Margari et al. 2010 and Tzedakis et al. 2009 should be added there (and not only for the last glacial period). Martrat et al., 2007 also present data for the penultimate glacial period and Antarctic ice core records also provide information on DO variability, since they form the basis of the synthetic Greenland record.

L. 31: I don't think further records are needed to "confirm" DO cycles, but additional high-resolution records like the one you are providing are needed to better understand these cycles.

L. 197: What do you mean by age offsets?

Figure 5: Please provide information on how you calculated DO pacing.

L. 215: I am not sure that's the case for the periods of interest and note that warmer conditions in SO would be expected with a weaker AMOC

L. 224: Please replace "confirms" by "provides additional evidence"

L. 227-228: "This could be related" iof "most likely"

Reviewer #2 (Remarks to the Author):

Review of Revised version: Dansgaard-Oeschger cycles of the penultimate and last glacial period recorded in stalagmites from Turkey by Held et al.

In this revised version of their manuscript, the authors have removed many of the difficult sections of the original version, which I commend. In particular, the authors now suggest that the longer timescale of DO variability during MIS 6, was due to a weaker and shallower mode of the AMOC. However, while this may have been the case, we have no direct evidence for the state of the AMOC during MIS 6 (in fact the state of AMOC during the LGM is still a matter of strong debate). The evidence cited here is all circumstantial, based on benthic oxygen isotopes, surface temperature and CO₂ records, and not proxies of AMOC strength or geometry. I therefore think that the authors have

to tone down this argument. Perhaps they could argue that their results (longer timescale of DO variability) might reflect a difference in the underlying structure and vigor of the AMOC during MIS 6, but that this conclusion awaits future direct and quantitative reconstructions of AMOC during that interval.

Other than this point, I don't have major concerns about the ms as presented.

Reviewer #3 (Remarks to the Author):

The revised manuscript and rebuttal letter addresses most of the comments and suggestions of the three reviewers and I very much acknowledge the effort of the authors to improve the manuscript. Particularly the temperature calibration approach has been removed from the manuscript together with the associated discussion.

Two issues raised in the first instance have not been properly addressed for my opinion.

First, more a technical aspect but mentioned by all reviewers, was to stipulate which data were already published in previous publications. In the rebuttal letter this has been made more or less clear, but it is not yet satisfactorily addressed in the manuscript itself.

Second, I cannot follow the authors response on the stacking procedure of their TWO partly very different MIS6 d13C records (n=2!). Since the records are partly very different, the stacking clearly affects the D-O assignment and interpretation. The mentioning of SPECMAP (n=5) and LR04 stack (n=57) is not very helpful here because they come along with a much more robust statistics! An average of 2 is problematic here.

These and some additional minor comments are included in the attached annotated pdf of the resubmitted paper.

Responses to Reviewers

We would like to thank all reviewers for their thoughtful and constructive remarks, which helped to further improve the quality of our manuscript. Please find our responses (in black) to the reviewer comments (in blue).

Reviewer: 1

Abstract, L. 12-15: A weaker AMOC as the reason behind the longer DO pacing is a hypothesis that cannot be tested here. I thus suggest to carefully phrase it as such (and also L. 227-230). In addition, here “background conditions” are also mentioned whereas they are not mentioned L. 227-230 (unless I misunderstood what you meant by background conditions).

We toned down the statement. We changed the term “background conditions” to “mean climate” and refer to it again in the discussion L. 225.

L. 22: There is a lot of evidence for DO cycles being associated with AMOC changes. What caused these AMOC changes is unclear, and volcanic forcing is one of the hypothesis being put recently forward but has only limited evidence, but not solar cycles.

Unfortunately, it is not clear to us how we can address this comment.

L. 23-24: I understand there are few terrestrial records of the penultimate glacial period which present DO variability but since they look at pollen, Margari et al. 2010 and Tzedakis et al. 2009 should be added there (and not only for the last glacial period). Martrat et al., 2007 also present data for the penultimate glacial period and Antarctic ice core records also provide information on DO variability, since they form the basis of the synthetic Greenland record.

We thank reviewer 1 for this comment, we added the references.

L. 31: I don't think further records are needed to “confirm” DO cycles, but additional high-resolution records like the one you are providing are needed to better understand these cycles.

Changed.

L. 197: What do you mean by age offsets?

We refer to "age offset" as the temporal differences in the chronology of D-O events between the Sofular, Sanbao, Huagapo and synth. ice core time series.

Figure 5: Please provide information on how you calculated DO pacing.

A detailed illustration of how the D-O pacing was determined is given in Supplementary Figures 6 and 7. This should be sufficient to document our approach to calculate the D-O pacing.

L. 215: I am not sure that's the case for the periods of interest and note that warmer conditions in SO would be expected with a weaker AMOC.

We modified the sentences to “...greater sea ice cover and lower amplitude SST variability ...” to avoid misunderstanding.

L. 224: Please replace “confirms” by “provides additional evidence”

Changed.

L. 227-228: “This could be related” iof “most likely”

Changed.

Reviewer: 2

In this revised version of their manuscript, the authors have removed many of the difficult sections of the original version, which I commend. In particular, the authors now suggest that the longer timescale of DO variability during MIS 6, was due to a weaker and shallower mode of the AMOC. However, while this may have been the case, we have no direct evidence for the state of the AMOC during MIS 6 (in fact the state of AMOC during the LGM is still a matter of strong debate). The evidence cited here is all circumstantial, based on benthic oxygen isotopes, surface temperature and CO₂ records, and not proxies of AMOC strength or geometry. I therefore think that the authors have to tone down this argument. Perhaps they could argue that their results (longer timescale of DO variability) might reflect a difference in the underlying structure and vigor of the AMOC during MIS 6, but that this conclusion awaits future direct and quantitative reconstructions of AMOC during that interval.

We thank reviewer 2 for this comment. We have toned down the statement.

Reviewer: 3

Two issues raised in the first instance have not been properly addressed for my opinion.

First, more a technical aspect but mentioned by all reviewers, was to stipulate which data were already published in previous publications. In the rebuttal letter this has been made more or less clear, but it is not yet satisfactorily addressed in the manuscript itself.

In the revised version, we provide detailed information on the use of previously published data and refer in more detail to the novelty of the data in this study.

Second, I cannot follow the authors response on the stacking procedure of their TWO partly very different MIS6 d13C records (n=2!). Since the records are partly very different, the stacking clearly affects the D-O assignment and interpretation. The mentioning of SPECMAP (n=5) and LR04 stack (n=57) is not very helpful here because they come along with a much more robust statistics! An average of 2 is problematic here.

We would like to emphasize that we are following common practice with this approach, and present further examples that also use this stacking method with a small number of records: Vollweiler et al. 2006 published in GRL (n=3), Bond et al. 2001 published in Science (n=4). Overlapping periods of stalagmite records of the combined Sanbao Cave timeseries (Cheng et al. 2016, published in Nature) are also represented by only 2 to a maximum of 3 records. Iscam is a frequently used program which, on a statistical basis, calculated a correlation coefficient for our record of 0.7 ($\delta^{13}\text{C}$) and 0.98 ($\delta^{18}\text{O}$), indicating that there is not that much difference between the two records. Last but not least, the records date back to more than 140,000 years before present and age offsets between the two records can be expected as dating uncertainties amount to several hundred years. Thus, it is not surprising that two records show small differences in their pattern as both are based on independent age models.

L. 2: Why not “Turkey”? Did the international english name changed?

Yes, the international name has changed to “Türkiye”.

Figure 2: Please use references to indicate data that are already published.

See first comment.

L. 109-120: Same as above. While you state in the rebuttal letter that you “have modified the introduction and methods to stipulate which data were already published in previous publications”, I feel this part can still be improved!

See first comment.

L. 125-127: Reference?

A reference is added in the revised version.

L. 130-132: This is not only the case for MIS 6 but particularly also for MIS 2-4!! The signal is not only dampened but the timing and sawtooth-like D-O pattern is affected.

We have changed this sentence. According to Fleitmann et al. (2009) and Corrick et al. (2020), the timing of D-O events during MIS 2-4 is not affected and is similar within the age uncertainties to other paleoclimate records discussed in these studies.

L. 138-141: I still find the stacking of only two records problematic for the MIS 6 interval (175-190) and mentioning of SPECMAP and LR04 stacks are not helpful here, since these examples deal with $n=5$ and $n=57$, respectively, coming along with a more robust significance. In your case, millennial-scale patterns in this interval are partly even anticorrelated.

See comment above.

Figure 4: Selection of correlation points is partly questionable (i.e. So 6.6 low temp. point, So 6.8 high temp. point, So 6.9 high temp. point...)

We have revised the correlation. Low temperature point of So 6.6 and high temperature point of So 6.8 have been changed. We do not recognize any alternative point for the high temperature point of So 6.8.

L. 183: See comment above.

Based on the context of our statement, it is unfortunately not clear to us which previous comment reviewer 3 is referring to. If it also refers to the stacking approach, please refer to the answer from the main comment at the beginning.

L. 186-188: Thank you for adding this data to suppl. Figure 2. In fact, So-57 shows an anticorrelated signal on the D-O scale. Negative $\delta^{13}\text{C}$ values correspond to cold/dry conditions in Lake Van. I think that the comparison is not as consistent as stated here.

We thank reviewer 3 for this comment. We modified this statement to “...broadly consistent...” to avoid misunderstandings. Considering the low resolution of the pollen record and the age uncertainties of both records (Pollen, Sofular), the overall trend and patterns of the pollen record and the Sofular $\delta^{13}\text{C}$ record are similar. We have modified Supplementary Figure 2 to emphasize the trending in both records.

L. 196: Please explain the difference between pacing and timing and how exactly you calculated the pacing.

The term "pacing" describes the time between D-O events, while "timing" refers to a certain point in time, such as the onset or peak of a D-O event. A detailed illustration of how the D-O pacing was calculated is given in Supplementary Figures 6 and 7.

L. 217: During peak (early) MIS 3, pacing is also closer to 4 ka, but associated rather with the warmest part of MIS 2-4. How does this observation fit to the proposed mechanism?

D-O variability is assumed to differ significantly between glacial and interglacials, thereby also having an effect on the threshold triggering D-O events. Our record shows an increase in $\delta^{13}\text{C}$ values to almost Holocene conditions for the early MIS 3. These climate conditions could have caused a change in the AMOC setting leading to a longer pacing.

REVIEWERS' COMMENTS

Reviewer #3 (Remarks to the Author):

I would like to thank the authors for taking the further comments of the reviewers into account. In their response letter, they address all raised issues and for my part, I have no further comments and look forward to the publication of this paper.